# Immunometabolic signatures predict risk of progression to sepsis in COVID-19

Ana Sofía Herrera-Van Oostdam[1], Julio E. Castañeda-Delgado[2,3], Juan José Oropeza-Valdez[1,3], Juan Carlos Borrego[4], Joel Monárrez-Espino[5], Jiamin Zheng[6], Rupasri Mandal[6], Lun Zhang[6], Elizabeth Soto-Guzmán[7], Julio César Fernández-Ruiz[1,3], Fátima Ochoa-González[3,8], Flor M. Trejo Medinilla[8], Jesús Adrián López[9], David S. Wishart[6], José A. Enciso-Moreno[3]*, Yamilé López-Hernández[2,10]*

1 Doctorado en Ciencias Biomédicas Básicas, Centro de Investigación en Ciencias de la Salud y Biomedicina, Universidad Autónoma de San Luis Potosí, San Luis Potosí, San Luis Potosí, México, 2 Cátedras-CONACyT, Consejo Nacional de Ciencia y Tecnología, Ciudad de México, México, 3 Unidad de Investigación Biomédica de Zacatecas, Instituto Mexicano del Seguro Social, Zacatecas, Zacatecas, México, 4 Departamento de Epidemiología, Hospital General de Zona #1 "Emilio Varela Luján", Instituto Mexicano del Seguro Social, Zacatecas, Zacatecas, México, 5 Christus Muguerza Hospital Chihuahua - University of Monterrey, Chihuahua, Chihuahua, Mexico, 6 The Metabolomics Innovation Center, University of Alberta, Edmonton, Alberta, Canada, 7 Maestría en Ciencias Biomédicas, Universidad Autónoma de Zacatecas, Zacatecas, Zacatecas, México, 8 Doctorado en Ciencias Básicas, Universidad Autónoma de Zacatecas, Zacatecas, Zacatecas, México, 9 MicroRNAs Laboratory, Academic Unit for Biological Sciences, Autonomous University of Zacatecas, Zacatecas, Zacatecas, Mexico, 10 Metabolomics and Proteomics Laboratory, Autonomous University of Zacatecas, Zacatecas, Zacatecas, Mexico

* ylopezher@conacyt.mx (YL-H); enciso_2000@yahoo.com (JAE-M)

## Abstract

Viral sepsis has been proposed as an accurate term to describe all multisystemic dysregulations and clinical findings in severe and critically ill COVID-19 patients. The adoption of this term may help the implementation of more accurate strategies of early diagnosis, prognosis, and in-hospital treatment. We accurately quantified 110 metabolites using targeted metabolomics, and 13 cytokines/chemokines in plasma samples of 121 COVID-19 patients with different levels of severity, and 37 non-COVID-19 individuals. Analyses revealed an integrated host-dependent dysregulation of inflammatory cytokines, neutrophil activation chemokines, glycolysis, mitochondrial metabolism, amino acid metabolism, polyamine synthesis, and lipid metabolism typical of sepsis processes distinctive of a mild disease. Dysregulated metabolites and cytokines/chemokines showed differential correlation patterns in mild and critically ill patients, indicating a crosstalk between metabolism and hyperinflammation. Using multivariate analysis, powerful models for diagnosis and prognosis of COVID-19 induced sepsis were generated, as well as for mortality prediction among septic patients. A metabolite panel made of kynurenine/tryptophan ratio, IL-6, LysoPC a C18:2, and phenylalanine discriminated non-COVID-19 from sepsis patients with an area under the curve (AUC (95%CI)) of 0.991 (0.986–0.995), with sensitivity of 0.978 (0.963–0.992) and specificity of 0.920 (0.890–0.949). The panel that included C10:2, IL-6, NLR, and C5 discriminated mild patients from sepsis patients with an AUC (95%CI) of 0.965 (0.952–0.977), with sensitivity of 0.993(0.984–1.000) and specificity of 0.851 (0.815–0.887). The panel with citric acid, LysoPC a C28:1, neutrophil-lymphocyte ratio (NLR) and kynurenine/tryptophan ratio discriminated severe patients from sepsis patients with an AUC (95%CI) of 0.829

**Data Availability Statement:** Raw data are available from the Mendeley database (doi: 10. 17632/x9tw3knwsd.1).

**Funding:** CONACyT grant number 311880 was awarded to Yamile Lopez Hernandez. The funder had no role in study design, data collection and analysis, decision to publish, or preparation of the manuscript.

**Competing interests:** The authors have declared that no competing interests exist.

(0.800–0.858), with sensitivity of 0.738 (0.695–0.781) and specificity of 0.781 (0.735–0.827). Septic patients who survived were different from those that did not survive with a model consisting of hippuric acid, along with the presence of Type II diabetes, with an AUC (95%CI) of 0.831 (0.788–0.874), with sensitivity of 0.765 (0.697–0.832) and specificity of 0.817 (0.770–0.865).

## Introduction

The novel coronavirus, SARS-CoV-2, was identified for the first time in December of 2019 in Wuhan, China. Coronavirus disease 2019 (COVID-19) is the third coronavirus infection that causes severe respiratory illness in humans [1]. The disease was declared a health emergency and a pandemic by the World Health Organization (WHO) causing more than 3 million deaths worldwide by April 2021 [2]. COVID-19 emerged as a complex disease with similar clinical characteristics of sepsis [3]. The WHO established definitions of pneumonia, acute respiratory distress syndrome (ARDS), and sepsis that apply to COVID-19 patients to characterize disease severity [4]. However, in clinical practice, it has been observed that many severe or critically ill COVID-19 patients develop clinical manifestations of shock (i.e. cold extremities, weak peripheral pulses, metabolic acidosis, and impaired liver and kidney functions) leading to the hypothesis that viral sepsis is crucial for COVID-19 and its associated mortality [5]. In fact, sepsis has been clearly documented in deceased patients in various cohorts [6–8].

The Third International Consensus Definitions Task Force [9] defined sepsis as a "life-threatening organ dysfunction caused by a dysregulated host response to infection", there by recognizing sepsis is a complex entity with both inflammatory and anti-inflammatory features. Many patients with severe COVID-19 meet this new definition (Sepsis-3). However, it has been described that the quick Sequential Organ Failure Assessment (qSOFA) score to clinically categorize a septic patient is not useful to identify COVID-19 patients with poor outcomes [10].

Proinflammatory cytokines are implicated in development of sepsis through to activation of inflammatory pathways by increasing the number, lifespan, and activity of innate immune cells [11]. Recent research also points to an important role of other factors, such as coagulation, microbiome composition, and thermoregulation [12], indicating that the pathogenesis of sepsis is influenced by alterations in the metabolic homeostasis. Therefore, the study of the host response (which involves both metabolic adaptations and the immune response), becomes crucial to understand the viral mechanism of sepsis. While cytokine-storm release syndrome (CRS) has been widely associated with sepsis, only a few studies have used a multiplex cytokine/chemokine profiling approach to predict sepsis [13]. Moreover, the integration of metabolomics and immune mediator's data to identify biomarkers predictive of coronavirus-associated has been even less explored. There is a need for a better understanding of the interaction between elevated inflammation signals, plasma metabolite composition, and immune cell dysfunction.

In the present work, 13 cytokines and chemokines involved in the antiviral response were measured in plasma samples of non-COVID-19 and COVID-19 patients with different disease severity, along with 110 metabolites quantified by liquid chromatography coupled to tandem mass spectrometry (LC–MS/MS) assay, to produce predictive panels of sepsis and mortality among COVID-19 patients.

## Materials and methods

### Patient enrollment and sample collection

Clinical data from 158 patients were collected retrospectively based on the epidemiologic survey done after admission to the Respiratory Triage of the Mexican Institute of Social Security from March to November 2020. Plasma samples were obtained at an early stage of the disease, namely, 4 days on average after symptoms onset and prior to laboratory diagnosis. A total of 37 individuals were confirmed negative (G1) for SARS-CoV-2 infection, who were suspected due to close contact with a confirmed case, and 121 had a confirmed infection by reverse-transcriptase polymerase chain reaction (RT-qPCR) from a nasopharyngeal swab specimen. From these, 41 were outpatients that developed a mild disease (G2), and 80 patients were hospitalized with different levels of severity according to the WHO disease severity scale [14]; 35 were classified as moderate/severe (G3), as these patients developed clinical symptoms of pneumonia or severe pneumonia with a moderate degree of hypoxia that required oxygen therapy) through face mask. The other 45 patients were included in the group of critical patients (G4), as they met at least one of the following criteria: Quick Sequential Organ Failure Assessment (qSOFA) $\geq$2 at the time admission, severe ARDS (PaO$_2$/FIO$_2$ $\leq$100mmHg), and reported life-threatening organ dysfunction during the illness (i.e. kidney and liver injury, and vascular and CNS complications). According to the Sepsis-3 definition, the presence of organ dysfunction is a key finding for the diagnosis of sepsis [9]. For these 120 infected patients, plasma samples were collected within 2 days of hospitalization, prior to antibiotic treatment if prescribed.

The qSOFA score (i.e. Glasgow coma scale <15, systolic blood pressure <100 mmHg, and respiratory rate >22 breath/min) obtained from the respiratory triage at emergency admission was computed resulting in 0, 1, 2, and 3 points based on the new sepsis consensus definition [9]. ARDS for mechanically ventilated patients was defined using the Berlin classification as mild (200 mm Hg < PaO$_2$/FIO$_2$ $\leq$300 mm) moderate (100 mm Hg < PaO$_2$/FIO$_2$ $\leq$200 mm Hg), and severe (PaO$_2$/FIO$_2$ $\leq$100 mm Hg) [14]. A description of the clinical features including demographic data, clinical symptoms and laboratory variables is provided in the Table 1.

The study was performed in accordance with the Declaration of Helsinki, and the Ethics Committee of the Mexican Institute for Social Security approved the study protocol (R-2020-785-068). Written informed consent was obtained from all participants. All patients included in this study were informed in writing regarding the collection of their samples for research aims and given the right to refuse such uses.

Blood cultures were performed only in those patients with clinical and distinctive signs of focal and/or bloodstream infections others than SARS-CoV-2 after >48 h post-hospitalization. Blood cultures were done with BD BACTEC Peds Plus™/F culture vials (Becton, Dickinson and Company, MD, USA) and BD BACTEC™ Plus Aerobic/F Culture Vials (Becton, Dickinson and Company, MD, USA) following manufacturer instructions.

### Metabolomics profile in plasma samples

Metabolites were measured using a locally developed LC-MS/MS metabolomics assay called The Metabolomics Innovation Centre (TMIC) Prime (TMIC PRIME®) Assay. The method, adapted to work with plasma using a similar quantitative assay developed for urine samples [15], provides quantitative results for up to 143 endogenous metabolites including biogenic amines, amino acids, organic acids, and lipid-like compounds.

Amino acids, biogenic amines and derivatives, and organic acids were analyzed by a reverse-phase LC-MS/MS custom assay, while glycerophospholipids, acylcarnitines, glucose and sphingomyelins were measured by Direct Infusion Tandem Mass Spectrometry

**Table 1. Sociodemographic, epidemiological, and clinical characteristics, including laboratory analyses, of the study participants by surveyed group.**

| Variables | G1 (N = 37) | G2 (N = 41) | G3 (N = 35) | G4 (N = 45) | p Value |
|---|---|---|---|---|---|
| Male sex, n (%) | 16 (43.2) | 23 (56) | 16 (45.7) | 29 (64.4) | 0.07 |
| Age, median years (Q1-Q3) | 41 (38–54) | 58 (51–64) | 53 (48–61) | 58 (46–64) | **0.0001**[a] |
| Smoking, n (%) | 3 (8.1) | 4 (9.7) | 3 (8.6) | 0 | 0.1 |
| Symptoms to sampling, median days (Q1-Q3) | 2 (1–5) | 3 (0–6) | 3 (1–6) | 5 (2–7) | **0.04**[b] |
| Symptoms before admission (Q1-Q3) | NA | NA | 4 (2–5) | 4 (2–6) | 0.8 |
| Systolic pressure ≤ 100, n (%) | NA | NA | 2 (5.7) | 6 (13) | 0.5 |
| Breathing frequency ≥ 22, n (%) | NA | NA | 15 (42.9) | 36 (80) | **0.0009** |
| Altered mental state GCS <15, n (%) | NA | NA | 2 (5.7) | 10 (22.2) | 0.06 |
| qSOFA, median (Q1-Q3) | NA | NA | 1 (0–1) | 1 (1–2) | **0.0001** |
| qSOFA 0 to 1, n (%) | NA | NA | 35 (100) | 33 (73.3) | **0.0008** |
| qSOFA ≥2, n (%) | NA | NA | 0 (0) | 12 (26.6) | **0.0008** |
| Pneumonia, n (%) | NA | NA | 2 (5.7) | 10 (22.2) | 0.06 |
| ARDS, n (%) | NA | NA | NA) | 34 (75.5) | - |
| PaO2/FIO$_2$, median (Q1-Q3) | NA | NA | NA | 65 (57–82) | - |
| Renal injury, n (%) | NA | NA | 0 (0) | 14 (30.4) | **0.0002** |
| Liver injury, n (%) | NA | NA | 0 (0) | 5 (11.1) | 0.06 |
| Organic vascular injury, n (%) | NA | NA | 0 (0) | 8 (17.7) | **0.008** |
| Central nervous system injury, n (%) | NA | NA | 0 (0) | 1 (2.2) | >0.9999 |
| Mechanical ventilation, n (%) | NA | NA | 0 (0) | 39 (86.6) | **<0.0001** |
| **Positive blood cultures after 48 h of hospitalization, n (%)** | NA | NA | 0(0) | 1(2.2) | >0.9999 |
| Death, n (%) | NA | NA | 8 (22.2) | 28 (77.7) | **<0.0001** |
| Days between hospital admission and death | NA | NA | 8(2–15) | 8(5–15) | 0.7 |
| Symptomatology, n (%) | | | | | |
| Fever | NA | 22 (53.7) | 21 (60) | 28 (62.2) | 0.4 |
| Cough | NA | 30 (73.1) | 27 (77.1) | 41 (91.1) | **0.03** |
| Headache | 27 (73.0) | 30 (73.1) | 21 (60) | 24 (53.3) | **0.03** |
| Dyspnea | 5 (13.5) | 13 (31.7) | 31 (88.6) | 32 (71.1) | **<0.0001** |
| Diarrhea | 2 (5.4) | 4 (9.8) | 6 (17.1) | 4 (8.8) | 0.5 |
| Chest tightness | 2 (5.4) | 6 (14.6) | 12 (34.3) | 8 (17.7) | 0.06 |
| Chills | 4 (10.8) | 14 (34.1) | 15 (42.8) | 10 (22.2) | 0.4 |
| Pharyngalgia | 14 (37.8) | 14 (34.1 | 15 (42.8) | 12 (26.6) | 0.4 |
| Myalgia | 14 (37.8) | 21 (51.2) | 19 (54.3) | 19 (42.2) | 0.7 |
| **Arthralgias** | 11 (29.7) | 22 (53.7) | 19 (54.3) | 18 (40) | 0.5 |
| **Rhinorrhea** | 6 (16.2) | 8 (19.5) | 6 (17.1) | 2 (4.4) | 0.09 |
| **Polypnea** | 1 (2.7) | NA | 6 (17.1) | 8 (17.7) | **0.04** |
| Anosmya | NA | 10 (24.4) | 7 (20) | 4 (8.8) | 0.05 |
| **Dysgeusia** | NA | 10 (24.4) | 7 (20) | 5 (11.1) | 0.06 |
| **Comorbidities, n (%)** | | | | | |
| **Diabetes (self-reported)** | 3 (8.1) | 4 (9.8) | 18 (51.4) | 9 (20) | **0.01** |
| **Obesity (>30Kg/m$^2$)** | 3 (8.1) | 8 (19.5) | 7 (20) | 13 (28.9) | **0.02** |
| **Hypertension (self-reported)** | 9 (24.3) | 11 (26.9) | 13 (37.1) | 17 (37.7) | 0.1 |
| **Admission Lab data, median (Q1-Q3)** | | | | | |
| **Erythrocytes (million/mL)** | 5.1 (4.8–5.5) | 5.2 (4.9–5.6) | 5.1 (4.9–5.4) | 5.1 (4.7–5.5) | 0.9 |
| **Hemoglobin (g/dL)** | 15.4 (14.7–16.3) | 15.3 (14.2–16.1) | 15.0 (14.4–15.8) | 15.3 (13.5–16.5) | 0.7 |
| **Platelets (thousands/ mL)** | 278.8 (238.0–327.0) | 257.0 (206.5–314.0) | 248.5 (213.0–274.0) | 243.0 (184.8–282.0) | 0.06 |
| **Leucocytes (×10$^3$)** | 7.1 (6.05–8.4) | 7.0 (5.4–8.3) | 8.6 (6.7–10.4) | 9.5 (7.6–12.1) | **0.0002**[c] |

*(Continued)*

**Table 1.** (Continued)

| Variables | G1 (N = 37) | G2 (N = 41) | G3 (N = 35) | G4 (N = 45) | p Value |
|---|---|---|---|---|---|
| Neutrophils (%) | 60.1 (54.5–66.0) | 66.6 (56.2–75.6) | 79.4 (75.3–83.0) | 85.4 (81.4–90.8) | **<0.0001**[d] |
| Lymphocytes (%) | 30.5 (25.8–36.0) | 25.1 (15.4–34.5) | 14.3 (10.6–16.8) | 8.8 (5.2–11.8) | **<0.0001**[e] |
| Neutrophils-Lymphocytes Ratio | 1.7 (1.5–2.2) | 3.0 (1.6–3.7) | 6.7 (4.5–7.4) | 11.2 (6.7–16.7) | **<0.0001**[d] |
| Monocytes (%) | 6.8 (5.3–8.7) | 7.1 (4.8–8.8) | 5.1 (3.0–6.1) | 3.5 (2.6–5.0) | **<0.0001**[d] |
| Glucose (mg/dL) | 93.0 (85.0–103.0) | 112.0 (95.8–125.5) | 134.3 (97.0–136.6) | 150.0 (113.0–247.0) | **<0.0001**[f] |
| Creatinine (mg/dL) | 0.9 (0.7–1.0) | 0.87 (0.7–1.0) | 0.85 (0.7–0.9) | 1.0 (0.8–1.5) | **0.01**[g] |

Continuous variables were compared using Mann-Whitney U tests or Kruskal-Wallis tests and categorical variables (sex, smoking, death, symptoms, and comorbidities) were compared using the chi-square test for trend, with p values of less than 0.05 considered statistically significant and shown in bold. The analyses were conducted using GraphPad Prism version 8.0.1 for Windows (GraphPad Software, La Jolla California USA.

[a] [G1 vs. G2; G1 vs. G3; G1 vs. G4];

[b] [G1 vs. G4];

[c] [G1 vs. G4; G2 vs. G4];

[d] [G1 vs. G3; G1 vs. G4; G2 vs. G3; G2 vs. G4];

[e] [G1 vs. G3; G1 vs. G4; G2 vs. G3; G2 vs. G4; G3 vs. G4];

[f] [G1 vs. G3; G1 vs. G4; G2 vs. G4]; and

[g] [G3 vs. G4].

(DI-MS/MS) analysis was performed on an ABSciex 4000 Qtrap tandem MS instrument (Applied Biosystems/MDS Analytical Technologies, Foster City, CA) equipped with an Agilent 1260 series UHPLC system (Agilent Technologies, Palo Alto, CA). The custom assay contained a 96-deep-well plate with a filter plate attached using sealing tape. Reagents and solvents were used to prepare the plate assay. The first 14 wells were used for one blank, three zero samples, seven standards, and three quality control samples.

## Sample preparation

For organic acid analysis, 150 μL of ice-cold methanol and 10 μL of isotope-labeled internal standard mixture [16] were added to 50 μL of plasma sample for overnight protein precipitation at –20˚C, followed by centrifugation at 13,000 × g for 20 minutes. A total of 50 μL of supernatant was loaded into the center of wells of a 96-deep-well plate followed by the addition of 3 nitrophenylhydrazine reagent. After incubation for 2 hours, butylated hydroxytoluene stabilizer (2 mg/mL) and water were added before LC-MS injection. For amino acids and biogenic amines and derivatives, glycerophospholipids, acylcarnitines, and sphingomyelins, samples were thawed on ice and subsequently vortexed and centrifuged at 13,000×g; 10 μL of each sample was then loaded onto the center of the filter on the upper 96-well plate and dried in a stream of nitrogen. Subsequently, phenyl-isothiocyanate was added for derivatization. After incubation, the filter spots were dried again using an evaporator. Extraction of the metabolites was then achieved by adding 300 μL of extraction solvent. The extracts were obtained by centrifugation into the lower 96-deep-well plate followed by a dilution step with the MS running solvent (0.2% formic acid in water, 0.2% formic acid in acetonitrile).

## LC-MS/MS method

An Agilent reversed phase Zorbax Eclipse XDB C18 column (3.0 mm × 100 mm, 3.5 μm particle size, 80 Å pore size) with a Phenomenex (Torrance, CA, USA) SecurityGuard C18 pre-column (4 mm × 3.0 mm) were used. The LC parameters used were as follows: mobile phase A

was 0.2% (v/v) formic acid in water, and mobile phase B was 0.2% (v/v) formic acid in acetonitrile. The gradient profile was as follows: t = 0 min, 0% B; t = 0.5 min, 0% B; t = 5.5 min, 95% B; t = 6.5 min, 95% B; t = 7.0 min, 0% B; and t = 9.5 min, 0% B. The column oven was set at 50˚C. The flow rate was 500 μL/min, and the sample injection volume was 10 μL. For the analysis of organic acids, the mobile phases used were A) 0.01% (v/v) formic acid in water, and B) 0.01% (v/v) formic acid in methanol. The gradient profile was as follows: t = 0 min, 30% B; t = 2.0 min, 50% B; t = 12.5 min, 95% B; t = 12.51 min, 100% B; t = 13.5 min, 100% B; t = 13.6 min, 30% B and finally maintained at 30% B for 4.4 min. The column oven was set to 40˚C. The flow rate was 300 μL/min, and the sample injection volume was 10 μL.

## DI-MS/MS method

The LC autosampler was connected directly to the MS ion source by red PEEK tubing. The mobile phase was prepared by mixing 60 μL of formic acid, 10 mL of water, and 290 mL of methanol; and the flow rate was programmed as follows: t = 0 min, 30 μL/min; t = 1.6 min, 30 μL/min; t = 2.4 min; 200 μL/min; t = 2.8 min, 200 μL/min; and t = 3.0 min, 30 μL/min. The sample injection volume was 20 μL.

## Quantification

To quantify organic acids, amino acids, and biogenic amines and derivatives, individual seven-point calibration curve was generated for each analyte. The ratios of each analyte's signal intensity to its corresponding isotope-labelled internal standard mixture were plotted against the specific known concentrations using quadratic regression with a $1/x^2$ weighting. Lipids, acylcarnitines, and glucose were analyzed semi-quantitatively. Single point calibration of a representative analyte was built, using the same group of compounds that share the same core structure, assuming linear regression through zero. All data analysis was done using Analyst 1.6.2 and MultiQuant 3.0.3.

## Cytokine and chemokine quantification in plasma samples

A premixed LEGENDplex™ Human Inflammation Panel (13-plex) (Biolegend, USA) was used to measure plasma cytokine and chemokine levels. The 13 cytokines and chemokines assayed simultaneously include IL-1β, IFN-α, IFN-γ, TNF-α, MCP-1 (CCL2), IL-6, IL-8 (CXCL8), IL-10, IL-12p70, IL-17A, IL-18, IL-23, and IL-33. Samples, reagents, and immunoassay procedures were prepared and performed according to the manufacturer's instructions. Briefly, plasma samples or standard mixture of the analytes were mixed with beads coated with capture on a 96 well filter plate for 2 hours. Beads were washed and incubated with biotin-labeled detection antibodies for 1 hour, followed by a final incubation with streptavidin-PE. Data was acquired using a FACS CANTO II flow cytometer (4-2-2 configuration, (BD Biosciences, USA) with FirePlex software. Analysis was performed using the LEGENDplex analysis software v8.0, which distinguishes between the 13 different analytes on basis of bead size and internal dye. The limit of detection (LOD) and the limit of quantitation (LOQ) were calculated from standard curves. All regression analyses showed an $R^2$ value >0.99.

## Statistical analysis

Medians (interquartile ranges [IQRs]) and frequencies (%) were used to describe baseline characteristics of non-COVID-19 subjects and patients for continuous and nominal data, respectively. Normality was assessed using the D'Agostino-Pearson normality test. Continuous variables were analyzed using Mann-Whitney U or Kruskal-Wallis tests. For nominal variables

(i.e. sex, smoking, death, symptoms, and comorbidities) chi-square tests for trends were used. P-values of less than 0.05 considered statistically significant. Analyses were conducted using GraphPad Prism version 8.0.1 for Windows (GraphPad Software, La Jolla California USA). Metabolites with more than 50% of missing values were excluded from statistical analysis. For metabolites with less than 50% of missing values, these were imputed by using half of the minimum concentration value for that metabolite. Log transformation, and auto-scaling were applied for data scaling and normalization. Univariate analysis of continuous and categorical data was performed by Mann–Whitney rank sum and Fisher's exact tests, respectively. Principal component analysis (PCA) and two-dimension partial least squares discriminant analysis (2-D PLS-DA) scores plots were used to compare plasma metabolite data across and between study groups; 2000-fold permutation tests were used to minimize the possibility that the observed separation of the PLS-DA was due to chance. Coefficient scores and least absolute shrinkage and selection operator (LASSO) algorithm were used to identify the most discriminating metabolites for group comparisons. Metabolite data analyses were done using MetaboAnalyst [16].

Metabolites with the highest VIP score and LASSO scores were used to create metabolite panels for sepsis COVID-19 status or outcomes using multivariate logistic regression. Receiver-operating characteristic (ROC) analyses were performed and sensitivity (Se), specificity (Sp) and the area under curve (AUC) with 95%CI were measured using MetaboAnalyst to identify the best metabolite/cytokine/chemokine combination panel predictive of COVID-19 group. In this analysis, balanced sub-sampling-based Monte Carlo cross validation (MCCV) was used to generate the ROC curves. Spearman correlations coefficients between concentration levels of metabolites and cytokines/chemokines were computed using the R package "corrr", and correlations plots were done using the "corrplot" package. Selected correlations for G2 and G4 were plotted in scatterplot using the package "ggplot2". Each analysis and plot were done in R studio (1.3.959).

Spearman correlations coefficients between concentration levels of metabolites and the disease severity (categorized as ordinal scale 1,2 and 3) was performed using GraphPad Prism version 8.0.1 for Windows (GraphPad Software, La Jolla California USA).

## Results

### Patients

Critically ill patients (G4) were more frequently male than those in the other groups. Comorbidities such as obesity, hypertension, and type II diabetes were more commonly found in patients that required hospitalization. Neutrophil: lymphocyte ratio (NLR) levels were higher among critically ill patients. Hospitalized patients (G3 and G4) showed pronounced lymphopenia and lower monocyte counts compared with non-hospitalized individuals (G2), but neutrophil counts were increased. The median time (IQR) from symptom onset to admission was 4 days (2–6). Three-quarters (34 patients) from the G4 group developed severe ARDS; two developed moderate ARDS, three died when intubated, and six developed hepatic, renal, and/ or vascular complications. In the G4 group, 39 patients were mechanically ventilated (86%); hospital mortality in this group was 60%. A statistically higher proportion of deaths was seen in G4 compared with G3. G4 patients manifested common characteristics of sepsis, including increased heart rate, respiratory failure, fever, leukopenia, hypotension and leukocytosis. Patients with the above complications meet the diagnostic criteria for sepsis [9]. Clinical and demographic characteristics comparing survivors versus non-survivors from the G4 group of patients, is presented in S1 Table.

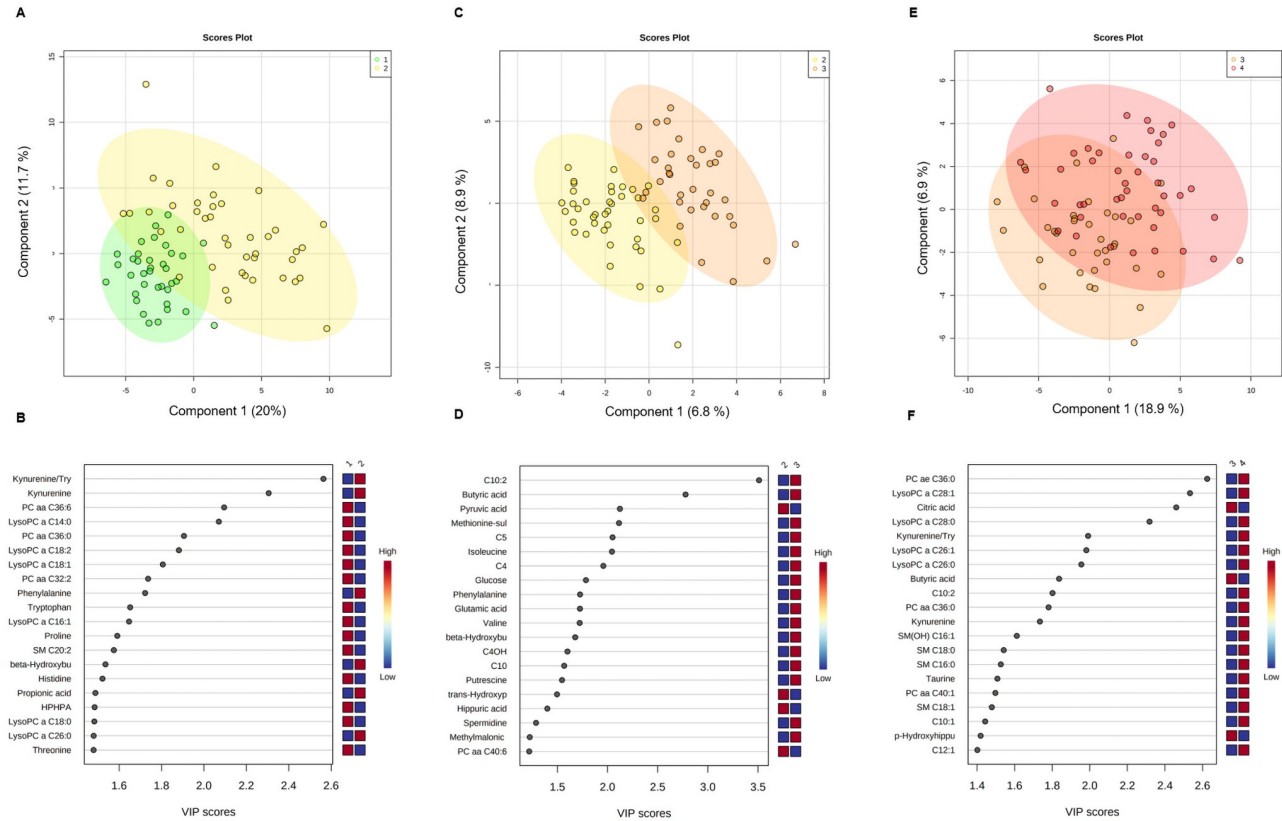

**Fig 1. Multivariate analyses from plasma metabolome profile of G1 versus G2, G2 versus G3, and G3 versus G4 patients.** (A) Score scatter plot based on PLS-DA models to explain the diagnosis (G1: green, G2: yellow) (B) rank of to 20 metabolites identified by PLS-DA according to VIP score on x-axis. (C) Score scatter plot based on PLS-DA models to explain the prognosis (G2: yellow, G3: orange), (D) rank of top 20 metabolites identified by the PLS-DA according to VIP score on x-axis. (E) Score scatter plot based on PLS-DA models to explain the prognosis (G3: orange, G4: red), (F) rank of top 20 metabolites identified by PLS-DA according to VIP score on x-axis. The most discriminating metabolites are shown in descending score order. The color boxes indicate whether metabolite concentration was increased (red) or decreased (blue). Figures were produced in MetaboAnalyst software v 4.0 (https://www.metaboanalyst.ca/).

## Metabolomics profile in COVID-19 patients

Differences in metabolite concentrations by disease severity were studied. For the comparison between G1 and G2, univariate analysis showed 30 metabolites significantly dysregulated after corrections for multiple comparisons (FDR < 0.05) including, kynurenine, kynurenine: tryptophan ratio, phenylalanine, propionic acid, beta hydroxybutyric acid, alpha-ketoglutarate, and alpha aminoadipic acid with higher concentrations in G2, while values for HPHPA, tryptophan, p-hydroxyhippuric acid, proline, histidine, threonine, citrulline, and lysine were higher in G1. Among lipids, one acylcarnitine (C10:1) and one lysophosphatydilcholine (LysoPC 26:0) were upregulated in G2, but PC aa 36:0, PC aa 32:2, PC aa 36:6, LysoPC a 14:0, LysoPC a 16:0, LysoPC a 16:1, LysoPC a 17:0, LysoPC a 18:0, LysoPC a 18:1, LysoPC a 18:2 and SM C (20:2) were so in G1. Multivariate analysis distinguished well between both groups (Fig 1A) with good performance parameters (accuracy:0.90, R2:0.85, Q2: 0.54). Variable importance in projection plots (Fig 1B) displays the 20 most dysregulated metabolites between G2 and G1. The metabolic pattern associated with disease severity was investigated by comparing outpatients with mild disease with hospitalized patients with moderate/severe illness (G2 vs. G3). Seven metabolites were dysregulated after corrections for multiple comparisons (FDR < 0.05). Among lipid species, three acylcarnitines (C10:2, C5, C4), were upregulated in

**Table 2. Plasma concentration values (micromoles) of dysregulated metabolites according to disease severity.**

| Metabolites | G1 N = 37 | G2 N = 41 | G3 N = 35 | G4 N = 45 | Correlation with disease severity (Spearman coefficient (r), p value (p)) |
|---|---|---|---|---|---|
| Kynurenine:Tryptophan | 0.034±0.007 | 0.08 (0.05–0.11) | 0.075 (0.05–0.12) | 0.13 (0.08–0.19) | 0.5, <0.0001 |
| Glutamic acid | 129.0 (104.0–157.0) | 143.0 (109.5–202.0) | 196.0±69.6 | 201.5 (134.5–243.8) | 0.4, <0.0001 |
| Phenylalanine | 61.05 (54.33–68.8) | 76.10 (61.45–91.70) | 83.9 (70.23–102) | 94.15±27.37 | 0.5, <0.0001 |
| Kynurenine | 1.76 (1.62–2.14) | 2.86 (2.22–4.14) | 3.09±1.62 | 4.12 (2.64–5.47) | 0.4, <0.0001 |
| Butyric acid | 0.67±0.24 | 0.75 (0.55–0.94) | 1.02 (0.89–1.28) | 0.89 (0.72–1.17) | 0.4, <0.0001 |
| Pyruvic acid | 101.6±23.2 | 132 (107.5–230.5) | 107.9±25.93 | 108 (82.9–132) | 0.02, 0.8 |
| LysoPC a C26:1 | 0.06 (0.05–0.09) | 0.081±0.03 | 0.09±0.03 | 0.11±0.03 | 0.5, <0.0001 |
| LysoPC a C26:0 | 0.12 (0.091–0.17) | 0.17±0.052 | 0.18±0.059 | 0.20 (0.17–0.24) | 0.4, <0.0001 |
| LysoPC a C28:1 | 0.14±0.043 | 0.14 (0.11–0.19) | 0.13 (0.12–0.20) | 0.19±0.065 | 0.3, <0.0001 |
| LysoPC a C28:0 | 0.16±0.039 | 0.19±0.060 | 0.19±0.067 | 0.23±0.069 | 0.3, <0.0001 |
| PC ae C36:0 | 1.28±0.28 | 1.24±0.28 | 1.37±0.35 | 1.72±0.53 | 0.3, <0.0001 |
| C4 | 0.18 (0.15–0.29) | 0.20 (0.14–0.28) | 0.23 (0.19–0.29) | 0.38 (0.19–0.51) | 0.2, 0.003 |
| C5 | 0.09 (0.07–0.14) | 0.10 (0.08–0.13) | 0.19 (0.13–0.26) | 0.22 (0.15–0.33) | 0.5, <0.0001 |
| C10:2 | 0.089 (0.07–0.10) | 0.10 (0.08–0.13) | 0.16±0.032 | 0.17±0.026 | 0.7, <0.0001 |
| C10:1 | 0.18 (0.039) | 0.21 (0.18–0.27) | 0.22 (0.18–0.29) | 0.25 (0.21–0.35) | 0.4, <0.0001 |
| C10 | 0.17 (0.14–0.24) | 0.223±0.095 | 0.26 (0.120–0.35) | 0.29 (0.19–0.38) | 0.4, <0.0001 |
| trans-Hydroxyproline | 11.5 (6.21–17.85) | 6.52 (3.6–10.04) | 4.54 (3.15–7.12) | 4.55±1.73 | -0.4, <0.0001 |
| Aspartic acid | 18.5 (13.7–26.2) | 15 (10.55–19.6) | 12.59±4.37 | 11 (7.57–16.4) | -0.3, <0.0001 |
| Tryptophan | 50.54±12.35 | 39.63±13.58 | 36.98±13.57 | 34.46±15.45 | -0.4, <0.0001 |
| Citric acid | 128.8±29.22 | 112 (84–136.5) | 100.2±29.7 | 81.9±26.09 | -0.5, <0.0001 |
| LysoPC a C14:0 | 4.12±1.62 | 1.78 (1.25–3.111) | 1.98 (1.51–2.34) | 1.83 (1.55–2.44) | -0.4, <0.0001 |
| LysoPC a C16:1 | 3.63 (2.77–4.25) | 2.34 (1.59–3.35) | 2.15±0.84 | 1.98 (1.59–2.57) | -0.3, <0.0001 |
| LysoPC a C18:2 | 25.09 (19.3–30.9) | 15.9 (9.14–22.5) | 12.71 (8.31–17.0) | 11.18 (7.77–13.9) | -0.5, <0.0001 |
| LysoPC a C18:1 | 20.95 (17.5–23.7) | 14.9 (11.0–20.6) | 14.9 (11.5–19.06) | 14.62±4.65 | -0.3, <0.0001 |

G3. Among amino acids, methionine-sulfoxide, and isoleucine were upregulated in G3. Butyric acid was also upregulated in G3, but pyruvic acid was downregulated. Multivariate analysis distinguished between G2 and G3 patients (Fig 1C) with good performance parameters (accuracy:0.96, R2:0.86, Q2: 0.70). VIP plots (Fig 1D) display the 20 most dysregulated metabolites between G2 and G3 patients. For the comparison between G3 and G4 patients, 10 metabolites were dysregulated (p <0.05), but none of them remained significant after corrections for multiple comparisons (FDR < 0.05). Among lipid metabolites, LysoPC a 26:0, LysoPC a 26:1, LysoPC a 28:0, LysoPC a 28:1, PC aa 36:0, PC ae 36:0 and C10:2 were elevated in G4 patients (p < 0.05). Kynurenine and the kynurenine:tryptophan ratio were also elevated in G4 patients, but butyric and citric acid decreased in this group. Multivariate analysis partially differentiated between both groups (Fig 1E). VIP plots (Fig 1F) show the 20 most dysregulated metabolites between G3 and G4.

To further understand how disease severity influenced the abundance of some metabolites, we evaluated the relationship among disease severity and the concentrations of metabolites. For this, a correlation analysis was performed. A very clear pattern emerged showing significant correlations (P<0.05) in 22 metabolites. As shown in Table 2 the plasmatic concentrations for metabolites are displayed for upward or downward trends with disease severity. The Spearman correlation coefficients (r) and p values are shown for this purpose.

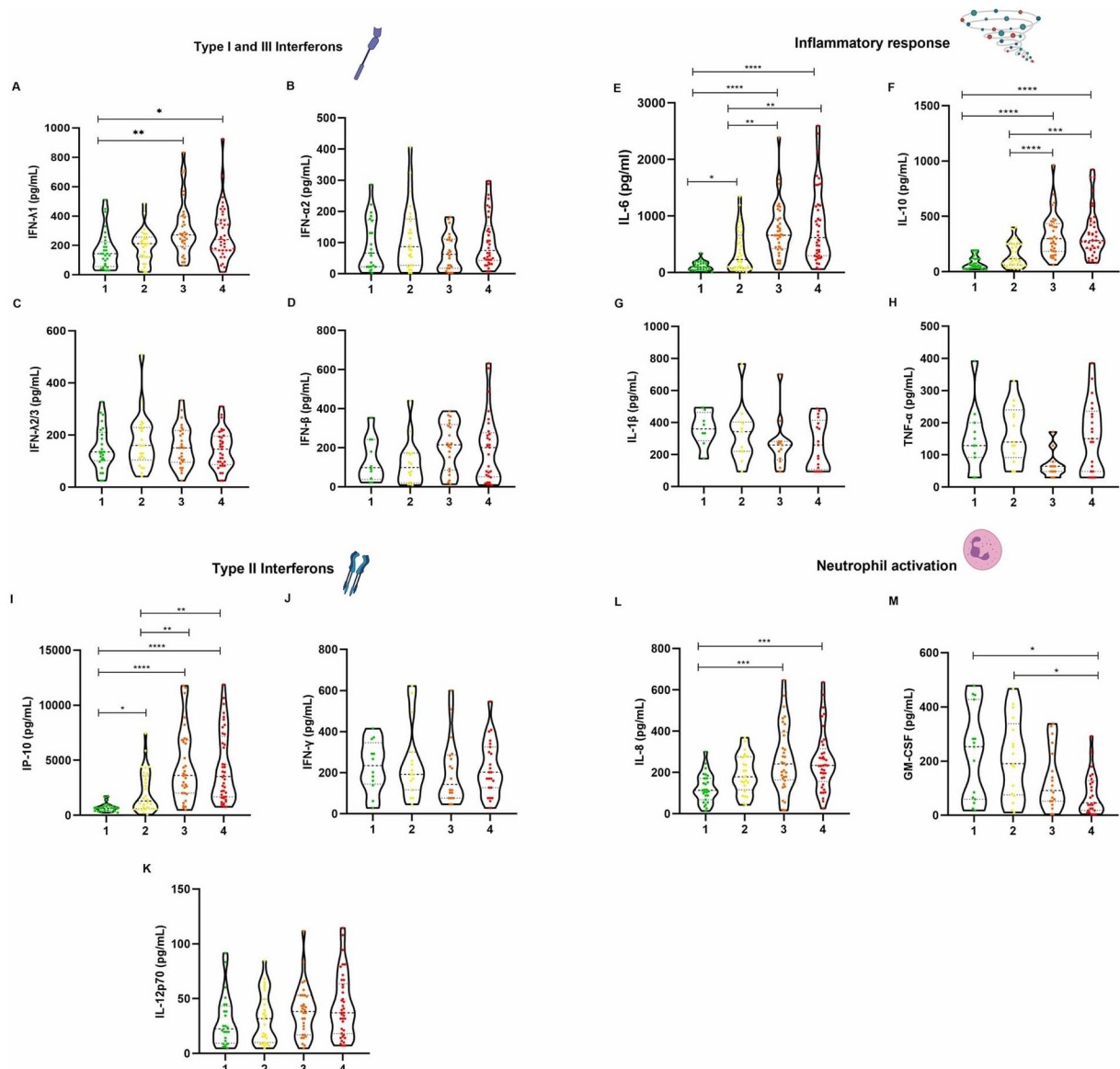

**Fig 2. Plasma concentration of types I, II and III interferon response cytokines, inflammatory response cytokines, and neutrophil response cytokines.** (A) IFN-λ1 (B) IFN-α2(C) IFN-λ2/3 (D) IFN-β (E) IL-6(F) IL-10 (G) IL-1β (H) TNF-α (I) IP-10 (J) IFN-γ (K) IL-12p70 (L) IL-8 (M) GM-CSF. All measurements were made using a multiplex flow cytometry assay (LEGENDplex). Results were obtained in a BD FACS Canto II flow cytometer and processed using the LEGENDplex Data Analysis Software v8.0. Graphs were constructed in GraphPad Prism v8.0. The * p value <0.05, ** p value <0.01, *** p value <0.001 and **** p value <0.0001 was calculated using Kruskall Wallis tests with a Dunn´s post-tests.

## Cytokines and chemokines linked to antiviral response are differentially produced in COVID-19 patients and are associated with disease severity

In order to analyze the immune regulatory circuits activated with antiviral immune responses, we measured the cytokine and chemokine responses in the plasma of non-COVID 19 and COVID 19 patients (Fig 2). Several immune mediators were analyzed, including Type I, II and III interferons, inflammatory cytokines and cytokines chemokines associated with/ neutrophil migration and differentiation. No differences were identified for Type I IFN α and β. Among

type III IFNs, only IFNλ-1 showed differences among study groups. Also, no differences were seen for IFN- γ and IL-12p70. However, a marked increase with severity was seen for IP-10 (CXCL10), a chemokine that is produced in response to IFN- γ activation. The cytokine release storm (CRS)-related, IL-6, IL-10, and IL-18, displayed a gradual increase with severity from G1 to all other three groups. Among cytokines regulating inflammatory responses, no differences were observed for IL-1β and TNF-α. However, IL-6 and IL-10 showed a marked increase by COVID-19 severity. IL-8 was also significantly increased in G3 and G4 patients compared with G1 individuals. For GM-CSF, differences also observed between the G1 and the other groups, yet with decreasing GM-CSF concentration as COVID-19 severity increased.

## Correlation between metabolites and cytokines profile showed a host response dysregulation associated with disease severity

Differences between G2 individuals (i.e. those that had well-coordinated response) and G4 patients (i.e. dysregulated response) were assessed using correlation matrices of metabolites and cytokines/chemokines. Fig 3 shows the pattern of correlations between metabolites/lipids and cytokines in G2 and G4, displaying only metabolites/lipids and cytokines/chemokines with correlation coefficients above 0.35 and $p < 0.05$. Overall, 6 positive and 6 negative significant correlations were found between metabolites and cytokines/chemokines and NLR in G2 individuals, but 10 positive and 2 negative significant correlations were seen in G4 patients, pointing to a stronger cross-talking between metabolic and immune mediators in severe forms of the disease. S1 and S2 Figs shows more detailed comparison between metabolites and lipids respectively, with cytokines/chemokines, in G2 and G4 patients.

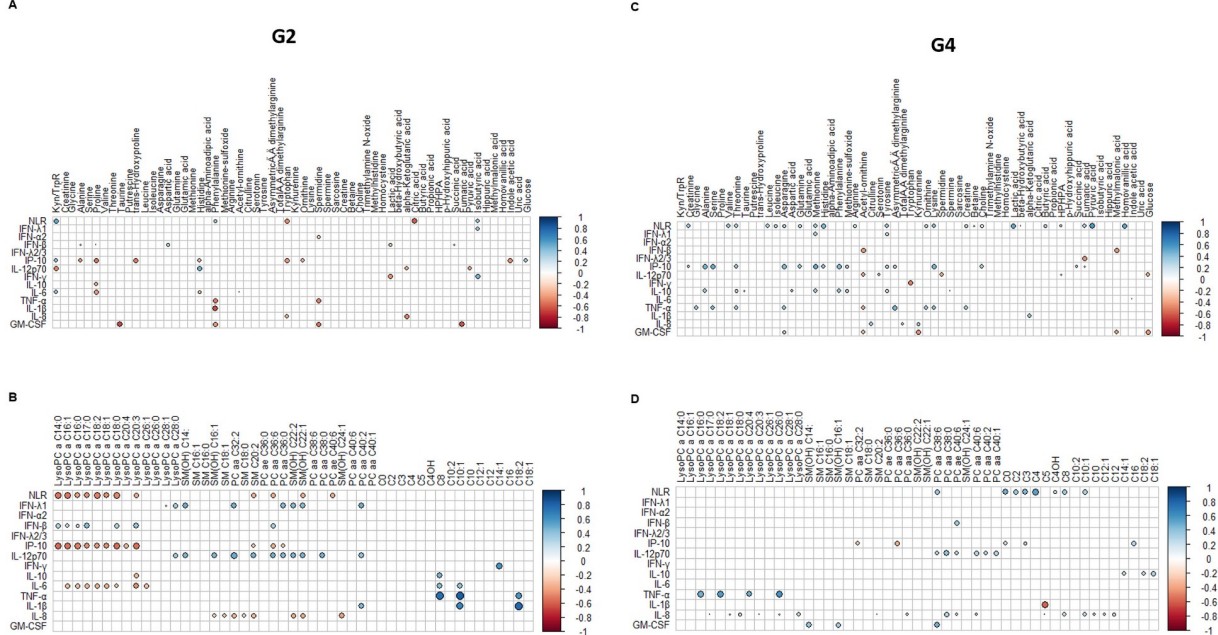

**Fig 3. Patterns of correlations among all metabolites and cytokines/chemokines measured in the study.** (A) Correlation between metabolites and cytokines/chemokines/ neutrophil to lymphocyte ratio (NLR) in mild patients (G2). (B) correlation between lipids and cytokines/chemokines/NLR in critically ill patients (G2). (C) correlation between metabolites and cytokines/chemokines/NLR in mild patients (G4). (D) correlation between lipids and cytokines/chemokines/NLR in critically ill patients (G4). The correlations between the concentration levels of the metabolites and cytokines and chemokines were done by Spearman's Correlation Coefficient using the R package "corrr", correlations plots were done using the "corrplot" package. Each analysis and plot were done in R studio (1.3.959).

## Logistic regression models to predict sepsis and in-hospital mortality caused by sepsis

Logistic regression models were built by combining metabolic and immune mediators to predict a) the occurrence of sepsis associated with SARS-CoV-2 infection, and b) the mortality in COVID-19 sepsis patients. Models were built using PLS-DA and LASSO scores, and metabolites/cytokines/chemokines plus clinical variables or co-morbidities with significant statistical differences across disease severity. When comparing G1 controls with G4 patients (considered as sepsis patients), a panel (Model A) made of kynurenine: tryptophan (OR = 5.59, p = 0.039, SE = 0.69), IL-6 (OR = 6.91, p = 0.019, SE = 0.82), LysoPC a C18:2 (OR = 0.06, p = 0.022, SE = 1.23), and phenylalanine (OR = 7.63, p = 0.019, SE = 0.83) was built. The AUC and 95% CI for the training/discovery set was 0.991 (0.986–0.995), the Se was 0.978 (95% CI 0.963–0.992) and the Sp was 0.92 (0.89–0.949). For 10-fold cross validation, the AUC was 0.967 (0.923–1.000), Se was 0.978 (0.978–1.000) and Sp was 0.917 (0.826–1.000). The equation for this model was: logit(P) = log (P / (1—P)) = 1.694 + 1.721 kynurenine:tryptophan + 1.933 IL-6 (pg/mL) - 2.838 LysoPC a C18:2 + 2.033 phenylalanine, where the numeric value of each metabolite in the equation was the concentration after log transformation and auto-scaling (Fig 4A).

When comparing G2 with G4 patients, the panel (Model B) made of C10:2 (OR = 74.26, p<0.001, SE = 1.14), IL-6 (OR = 2.58, p = 0.0329, SE = 0.44), NLR (OR = 2.73, p = 0.021, SE = 0.43), and C5 (OR = 2.36, p = 0.048, SE = 0.43) was produced. The AUC and 95% CI for the training/discovery set was 0.965 (0.952–0.977), with Se of 0.993 (0.984–1.000) and Sp of 0.851 (0.815–0.887). For 10-fold cross validation, the AUC was 0.941 (0.886–0.997), Se of 0.978 (0.978–1.000) and Sp of 0.854 (0.745–0.962). The equation for this model was: logit(P) = log (P / (1—P)) = 0.239 + 4.308 C10:2 + 0.949 IL-6 (pg/mL) + 1.005 NLR + 0.86 C5, where the numeric value of each metabolite in the equation was the concentration after log transformation and auto-scaling (Fig 4B).

When comparing G3 with G4 patients, a panel (Model C) with citric acid (OR = 0.43, p = 0.009, SE = 0.31), LysoPC a C28:1 (OR = 2.12, p = 0.008, SE = 0.28), NLR (OR = 1.66, p = 0.06, SE = 0.27), and kynurenine:tryptophan ratio (OR = 1.78, p = 0.038, SE = 0.27) was produced. The AUC and 95% CI for the training/discovery set was 0.829 (0.800–0.858), with Se of 0.738 (0.695–0.781) and Sp of 0.781 (0.735–0.827). For 10-fold cross validation, the AUC was 0.785 (0.685–0.884), Se = 0.733 (0.733–0.863) and Sp = 0.771 (0.632–0.911). The equation for this model was: logit(P) = log (P / (1—P)) = 0.385–0.833 citric acid + 0.753 LysoPC a C28:1 + 0.509 NLR +0.576 kynurenine:tryptophan, where the numeric value of each metabolite in the equation was the concentration after log transformation and auto-scaling (Fig 4C).

The clinical parameter to assess the patients' severity at the emergency unit was the qSOFA. The model built with qSOFA (Fig 4D) showed an AUC of 0.74 (95% CI 0.66–0.82) (p = 3.1286E-5), which was slightly lower than the logistic Model C (Fig 4C) built with metabolites and NLR.

When G4 survivors (n = 17) were compared with non-survivors (n = 28), PLS-DA showed a partial separation between both groups (S3A Fig). VIP scores identified the metabolites responsible for the separation between both groups (S3B Fig). The panel (Model D) shown in Fig 4E included hippuric acid (OR = 0.36, p = 0.03, SE = 0.47), and Type II Diabetes (OR = 2.67, p = 0.05, SE = 0.51. The AUC and 95% CI for the training/discovery set was 0.831 (0.788–0.874), with Se of 0.765 (0.697–0.832) and Sp of 0.817 (0.770–0.865). For 10-fold cross validation, the AUC was 0.813 (95% CI 0.669–0.957), Se of 0.765 (0.765–0.966) and Sp of 0.821 (0.680–0.963). The equation for this model was: logit(P) = log (P / (1—P)) = -0.821 + 0.983

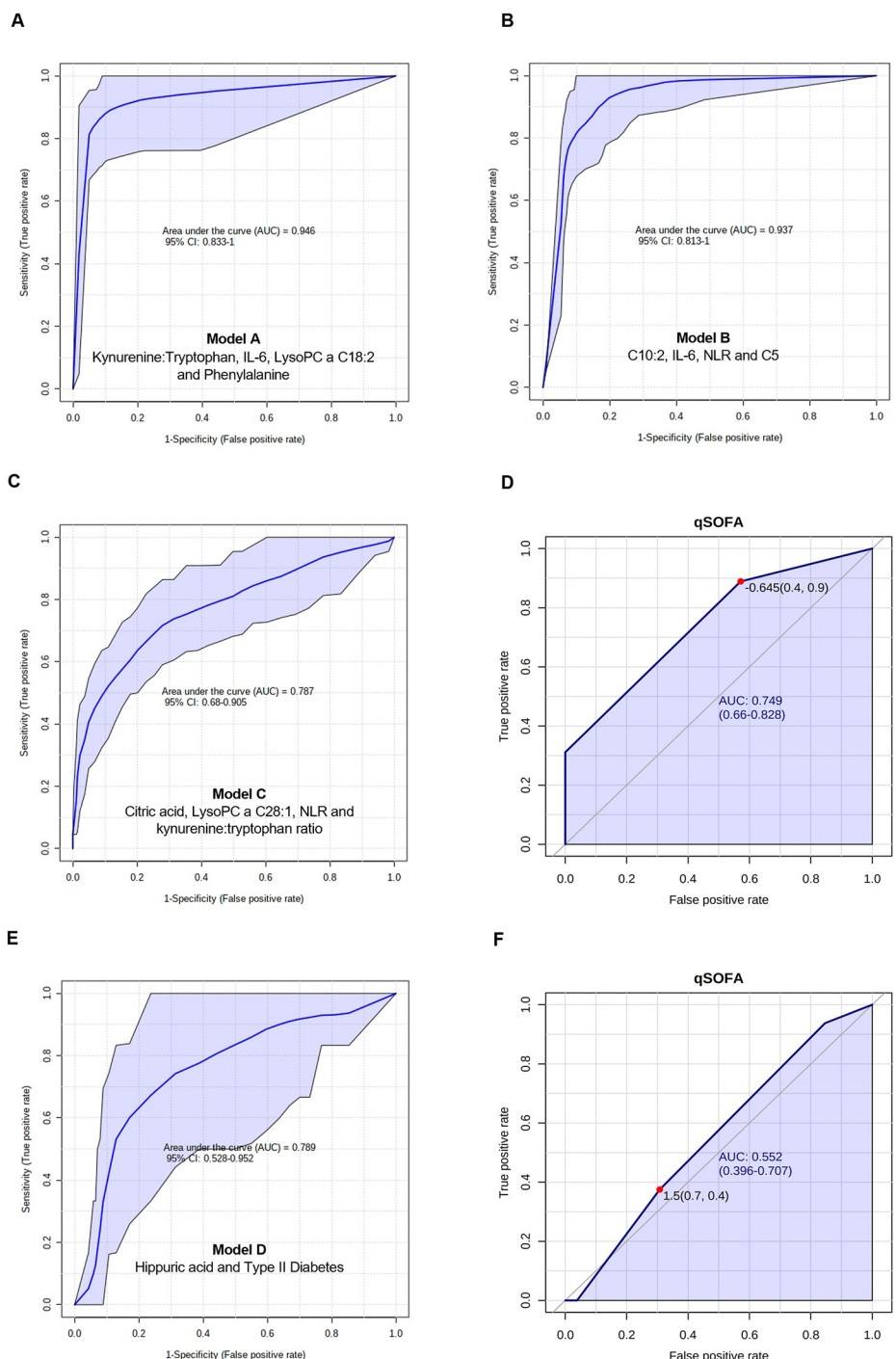

**Fig 4. Multivariate and univariate models.** (A) Multivariate ROC curve of model A: G1 vs. G4. (B) ROC curve of model B: G2 vs. G4. (C) ROC curve of the model C: G3 vs. G4. (D) Univariate ROC curve explaining the performance of qSOFA to predict sepsis in G3 and G4. (E) Multivariate ROC curve of model D: non-survivors vs. survivors from G4. (F) Univariate ROC curve explaining the performance of qSOFA to predict mortality in G4. Figures were made in MetaboAnalyst software v 4.0 (https://www.metaboanalyst.ca/).

Type II DM—1.032 hippuric acid. This model showed a better performance than qSOFA to predict mortality among G4 patients, with an AUC of 0.55 (95% CI 0.39–0.70) (Fig 4F).

Finally, the proposed models were also adjusted for obesity and diabetes, to address the influence of these comorbidities in immune and metabolic differences found in patients with different disease severity grades. S2 Table shows the adjusted models considering diabetes and obesity. When comorbidities were considered by separate, obesity is an important risk factor to discriminate between G1 and G4, while diabetes is the important factor to discriminate between G3 and G4. However, none of these comorbidities significantly contribute to the logistic regression models built with the different panels of metabolites and cytokines. This was also confirmed by the analysis of metabolomic profile of COVID-19 patients with diabetes and obesity (58 patients) and without these comorbidities (53 patients). S4A Fig shows that in the PLS-DA there is no clear separation between COVID-19 patients with diabetes and obesity (COVID-DO) and COVID-19 patients without diabetes and obesity (COVID-WDO), also demonstrated by the 10-cross-validation (S4B Fig). In the univariate analysis, only four metabolites were significant after FDR correction: glucose, C4OH, TMAO, and lysoPC a 18:0 (S4C Fig).

## Discussion

This study assessed 110 plasma metabolites and 13 cytokines/chemokines implicated in the antiviral response of COVID-19 patients at various levels of disease severity with the ultimate goal of producing predictive panels of sepsis and mortality among critically ill patients. The plasma samples analyzed in the present study were collected within four days of symptoms onset and in the first 24 h after hospitalization. Patients were confirmed SARS-CoV-2 positive by qRT-PCR, demonstrating the presence of a viral infection. No other symptoms suggesting bacterial co-infections were detected, at least at the moment of blood sampling. Increased lymphopenia was observed in COVID-19 patients, implying possible damage of lymphocytes by the SARS-CoV-2 virus as has been previously hypothesized [17]. Since the patients arrived at the respiratory TRIAGE with a positive SARS-CoV-2 test, no blood cultures were indicated by protocol upon admission. Only four patients had blood cultures drawn at least two weeks after hospitalization (1.2% had a positive blood culture), being *Kleibsella oxytoca* the microorganism detected. This is a gram-negative bacterium, closely related to *Kleibsella pneumoniae*, causing nosocomial infections mainly in diabetic patients, critically ill patients or under antimicrobial therapies.

In line with our findings, a study from New York City hospitals reported very low rate of true bacteremia (1.6%) among COVID-19-positive patients [18]. In a study done by Yang et al. [19], bacteremia was seen in 3% cases among non-survivors of COVID-19 patients. In another study, among 267 patients hospitalized with COVID-19 pneumonia, 38 had early blood cultures drawn. No clinically relevant microorganism was isolated from blood and contaminant microorganisms were recovered in 18% of patients, suggesting no evidence of bacteremia in patients with COVID-19 pneumonia [20]. Drake et al [21] also reported higher rate of complications primarily driven by non-infectious complications, as the rates of secondary bacterial infection in patients with COVID-19 were lower than those described in influenza infection. The incidence of secondary pulmonary infections reported by Chong et al [22] was 16% for bacterial infections and lower for fungal infections (6.3%) in hospitalized COVID-19 patients.

The Global Sepsis Alliance has stated that SARS-CoV-2 causes sepsis (available at: https://www.global-sepsis-alliance.org/covid19), either directly (because is an infectious agent demonstrated by qRT-PCR), or indirectly, due to the damage to tissues, immune system, etc. The increased lymphopenia observed in COVID-19 patients may lead to secondary infections

**Table 3. Comparison of metabolic and immune mediators in sepsis caused by bacterial, fungal and viral infections.**

| ETIOLOGY | METABOLITES AND/OR CYTOKINES | REFERENCE |
|---|---|---|
| Sepsis (pulmonary infections) | glucose, citrate, glycine, histidine, 3-hydroxybutyrate, creatinine | [36] |
| SIRS or sepsis (*Streptococcus pneumoniae*, *Escherichia coli* and *Staphylococcus aureus*) | Cis-4-decenoylcarnitine, 2-methylbutyroylcarnitine, butyroylcarnitine, hexanoylcarnitine, lactate, age, and hematocrit | [37] |
| Community-acquired Pneumonia (CAP) in sepsis or severe sepsis/ septic shock | lysoPC a C26:1 | [38] |
| Bacteremic sepsis | myristic acid, citric acid, isoleucine, norleucine, pyruvic acid and a phosphocholine like derivative | [39] |
| Sepsis or septic shock (different causes) | isoleucine, alanine, acetylcarnitine, lactic acid, pyruvic acid, LysoPG (22:0) and LysoPC (24:0) | [40] |
| CAP | Kynurenate, urea, fumarate | [41] |
| Sepsis (AKI, ARDS) | leucine, glutamic acid, cysteine, methionine, phenylalanine, putrescine, and aspartic acid, serine, tryptophan, glutamine, d-Proline, (N-methoxycarbonyl-, octyl ester) and asparagine, lactic acid, adipic acid, and 3-hydroxypropionic acid, pyruvic acid and nicotinamide adenine dinucleotide phosphate (NADP)-NADPH | [42] |
| Sepsis caused by *P. aeruginosa* burn | *trans*-4-hydroxyproline, 5-oxoproline, glycerol-3-galactoside, indole-3-acetate, and indole-3-propionate | [26] |
| Bacteremic sepsis | (4E,8E,10E-d18:3) sphingosine | [43] |
| | Oleoylcarnitine/ elaidic carnitine | |
| | PC(O-18:0/0:0) | |
| | PC(19:0/0:0) | |
| | Arachidonic acid methyl ester | |
| Septic shock by different causes including bacterial pneumonia | isobutyrate, myo-inositol, proline, urea, 3-hydroxybutyrate, O-acetylcarnitine, 2-hydroxybutyrate phenylalanine. | [31] |
| | IP-10, HGF, IL-18, IL-1 and IL-1Ra, IL-2Ra. | |
| Bacterial sepsis (*Klebsiella pneumoniae, Staphylococcus aureus, Neisseria meningitidis, Pseudomonas aeruginosa, Stenotrophomonas maltophilia, Streptococcus pneumoniae, Streptococcus pyogenes*) And Viral sepsis (*Adenovirus, Coronavirus, Herpes Simplex Virus, Influenza A, Respiratory Syncytial Virus, Rhinovirus, Varicella Zoster Virus*) | bacterial infection from controls: myo-inositol, phenylalanine, lactate, pyruvate and 2-hydroxyisobutyrate | [44] |
| | viral infections from controls: 3-hydroxybutyrate, urea, valine 2-methylglutarate and isobutyrate. | |
| | Viral versus bacterial sepsis: 2-hydroxyisovalerate, alanine, citrate, creatine phosphate, creatinine, histidine, isoleucine, ornithine and tyrosine. | |
| Bacterial sepsis patients with pulmonary infection (Gram negative bacteria) and SARS-CoV-2 sepsis patients | IL-1β, IL-2R, IL-6, IL-8, IL-10, and tumor TNF-α) were observed in both bacterial sepsis and SARS-CoV-2 sepsis groups, but were lower in the latter group than in the former | [45] |
| COVID-19 critically ill patients with acute respiratory distress syndrome (ARDS) or sepsis due to other causes | There were no statistically significant differences in baseline levels of IL-1β, IL-1RA, IL-6, IL-8, IL-18, and TNF-α between patients with COVID-19 and critically ill controls with ARDS or sepsis | [46] |

during hospitalization [23]. We cannot rule out with the present work the existence of late bloodstream infections since we only sampled in the first 24 h after hospitalization, and consequently, the immunometabolic signatures reported by us belong to this first hospitalization hours. Invasive devices, diabetes, glucocorticoid treatment, and combination of antibiotics have been found to be significant predictors of nosocomial infections [24]. Intestinal damage due to SARS-CoV-2 infection, systemic inflammation-induced dysfunction, and IL-6-mediated diffuse vascular damage may increase intestinal permeability and precipitate bacterial translocation [25] enhancing the susceptibility to secondary pulmonary infections that are predominantly seen in critically ill hospitalized COVID-19 patients.

The plasma metabolic signature reported by us is consistent with a septic process, as it has been previously described for bacterial [26–28] or viral [29, 30] sepsis. As shown in Table 3, several works have examined the alterations in metabolite levels associated to sepsis or septic shock induced by different etiologic agents, for example bacteria and fungi. To date, no single

compound has shown sufficient sensitivity and specificity to be used as a routine biomarker for early diagnosis and prognosis of septic shock. In terms of immunometabolism, only a few works have been published [31, 32] demonstrating that a combination of metabolic and immune biomarkers, may improve the identification and the prognosis for sepsis. Validated markers to differentiate between viral or bacterial sepsis have not been developed up to date. In the Table 3 we can observe that, in general, metabolic pathways, such as glycolysis, TCA cycle, fatty acid oxidation, and amino acid pathways, play important roles in sepsis and septic shock associated with different causal agents. Regarding inflammatory cytokine levels, several authors are challenging the major role of cytokine storm in the disease pathogenesis [33], since the levels found in COVID-19 patients have been significantly lower than those reported for bacterial sepsis or ARDS [34, 35]. However, it is important to acknowledge that in the present work we found higher IL-6, IL-8, IL-1 β and TNF- α levels, compared and consistent to those commonly described for sepsis.

It has been described that the biochemical features distinctive of COVID-19 sepsis point to a state of acute inflammation with a cytokine storm associated with an altered metabolism [47] that modulates immune responses against infectious agents, increasing or decreasing the release of pro- and anti-inflammatory cytokines [48]. During this process, activated immune cells undergo extensive metabolic rewiring for energy and biomass generation whereby metabolites become important regulators of immunity and disease [49]. However, in the context of the COVID-19 pandemic, there is limited data exploring the interplay between metabolism and immune system [50–54]. Metabolic signatures showing alterations in plasma levels of kynurenine, phenylalanine, lysophosphatidylcholines species and acylcarnitines have already been reported in different settings of septic shock patients [37, 55–57] suggesting an overall derangement of energy circuits and lipid homeostasis as indicators of disease severity, as summarized in Fig 5.

An elevated concentration of phenylalanine appears to be the result of an accelerated rate of protein breakdown often caused by infections and inflammatory states [31, 58]. The metabolites referred here have been altered in sepsis patients [59], and are believed to have prognostic value for sepsis, especially amino acids and derivatives, lipids and lipid-like molecules, and organic acids and derivatives [40]. Lysophosphatydilcholines are bioactive lipids for their immune activation enrollment and are considered mediators during the development of sepsis [60]. Some studies reported that serum concentrations of LPC subtypes 16:0, 18:0, 18:1, and 18:2 were lower in septic patients compared with healthy controls [61, 62] suggesting that impaired metabolic homeostasis can cause the sustained low levels of LPC in septic patients [63]. One study reported that glycerophospholipid LysoPC a C26:1 discriminated well patients with community acquired pneumonia from those with other etiologies of sepsis with high Se and Sp [38]. A possible association between plasma acylcarnitine levels and organ dysfunction, systemic inflammation, and sepsis has also been reported in animal models studies [64]. Moreover, increased plasma levels of short- and medium-chain acylcarnitines have been associated with hepatobiliary dysfunction, renal dysfunction, thrombocytopenia, and hyperlactatemia in septic patients too [65]. The results of this study support the idea of the mitochondrial manipulation by the virus for replication purposes. Mitochondrial dysfunction, dysregulation in oxidative phosphorylation, and associated oxidative stress appear to drive the production of proinflammatory cytokines that ultimately play a key role in the immune response. Evidence indicates that mitochondrial dysfunction is essential for the induction and propagation of sepsis-induced organ injury, which has been demonstrated in both animal and human studies [66–69].

Among cytokines, IL-6 serves as an important mediator during the acute phase of response to inflammation in sepsis, and its clinical value has been assessed in patients with various septic

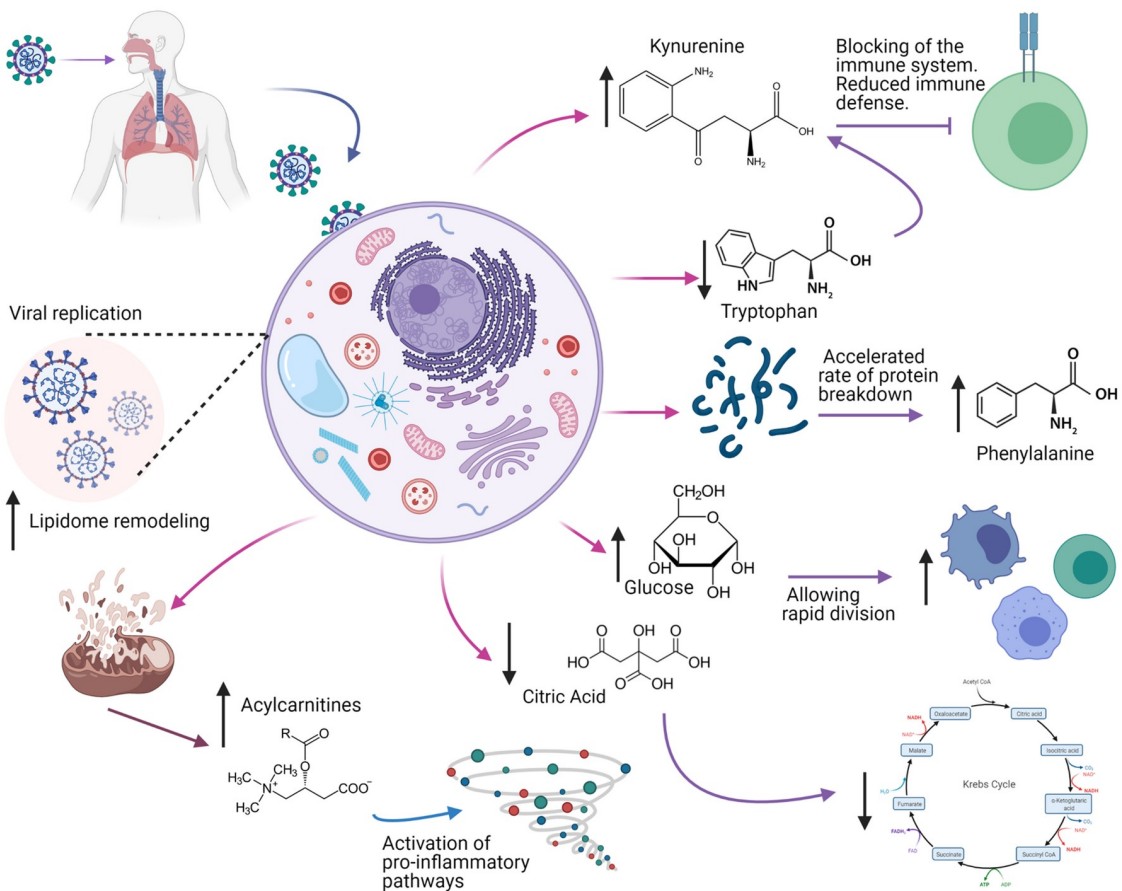

**Fig 5. Differential metabolites involved in sepsis associated to COVID-19 infection.** Black arrows represent the trend (increasing or decreasing) in critically ill patients. The figure was created with BioRender.com under a CC BY license, with permission from BioRender, original copyright 2021.

conditions in several studies [70–72]. Authors have reported that serum IL-6 levels discriminated sepsis (AUC 0.83–0.94, p < 0.001; cut-off value, 52.6 pg/mL, 80.4% Se, 88.9% Sp) from controls, and also distinguished septic shock (AUC 0.71–0.89; cut-off value, 348.92 pg/mL, 76.1% Se, 78.4% Sp) from sepsis [73]. In the context of COVID-19 epidemic, other researchers found that the maximal level of plasma IL-6 was higher in critical cases than in severe cases (median 1072.98 pg/mL (IQR 453.85–9163.64) vs. 49.14 pg/mL (IQR 27.98–101.30); p = 0.001) [74]. Such findings point to host sepsis as contributing to the extraordinary elevation of plasma IL-6 in critically ill patients with COVID-19.

Over the past decade, the field of immunometabolism has produced knowledge of the various mechanisms by which cells modulate metabolism to achieve effector functions necessary to fight infection and maintain homeostasis, and now it is accepted that all immune cells depend on specific and efficient metabolic pathways to perform an appropriate response [75]. The use of this knowledge is thus relevant for the understanding of COVID-19 pathophysiology given the complexity of the disease and the involvement of viral and host factors associated with the disease severity and outcome. The correlation analysis between metabolites and cytokines/chemokines in our study showed a significant crosstalk between metabolism and immune system principally in patients who developed a severe disease. Some metabolites involved in amino acid, energy, and lipid metabolisms, are implicated in the initiation of the

immune response. The dysregulated metabolites found have been previously associated with mTOR activation, which has a unique intracellular signaling position, and it is a critical regulator of the immune response by playing a central role in the sensing nutrient availability, cytokine/growth factor signaling, and costimulatory factors. The metabolic activities promoted by mTOR allow the accumulation of biomass necessary for cell division or activation. It is activated by nutrients (i.e. glucose, amino acids, and lipids), growth factors, insulin, and inflammatory cytokines [76].

The data presented here suggest that metabolic rewiring of the cells metabolic pathways may account for such observed changes and impaired functions. For this reason, the building of predictive models for sepsis, including not only metabolites, but also cytokines, chemokines and NLR can show higher performances. Considerably elevated NLR (>10) suggests the presence of severe systemic stress (as could be caused by septic shock) [77]. The results of ROC analysis presented here indicate the best sensitivity, specificity, accuracy and the highest AUROC values for the integrated metabolomics-cytokine/chemokine/NLR approach compared to the diagnostic and prognostic power of qSOFA scores. So far, their performance (Se and Sp) has not been adequate to predict sepsis [78]. Recently, it was reported that COVID-19 patients who developed ARDS had a mean qSOFA score of one on ICU admission, and no differences between those who were mechanically ventilated compared with those without ventilator support [79], leading to the suggestion that qSOFA was not appropriate to predict COVID-19 patient with poor outcomes typical of sepsis.

In the present work we also demonstrated that there is not differentiation between diabetic and obese COVID-19 patients from those COVID-19 patients without these comorbidities. Only four metabolites were detected as significantly altered between both groups: glucose, TMAO, lysoPC a 18:0 and C4OH. These metabolites have been previously associated with obesity and diabetes [80–82] and may contribute to the inflammatory state described in these patients. Other authors have also explored how the presence of these comorbidities influences the metabolomics differences found in COVID-19 patients concerning healthy controls and patients with different severity grades. Shi et al [83] compared the serum metabolome profiles of COVID-19 patients with and without comorbidities. The results showed that none of the groups of patients with hypertension, diabetes or fatty liver disease could be distinguished from patients without comorbidities. Similarly, Marin Corral et al [84] showed similar results in metabolic pathways when analyzing COVID-19 non-obese and obese patients separately. These results suggest that the effect of SARS-CoV-2 infection on the metabolome profile is much greater than pre-existing conditions such as diabetes and obesity.

## Conclusions

Our results integrate immune and metabolic signatures of COVID-19 associated sepsis and provide supporting evidence for the recognition of sepsis in patients who develop a critical disease. Our results also demonstrated the cross-talking between metabolic and immune response against the viral infection with SARS-CoV-2. The metabolic alterations observed in critically ill patients are in accordance with previously described alterations for sepsis/septic shock caused by viral or bacterial agents. Based on our integrative study, we propose the use of different panels for 1) early categorization of at risk patients, 2) prioritize care and 3) accurately assess prognosis of sepsis and mortality with high predictive values.

## Limitations of the study

One limitation is the cross-sectional exploratory nature of the study design. This design prevented a longitudinal metabolite and cytokine/chemokine assessment and so further

prospective observational designs, such a cohort with repeated measures would be needed to follow-up on the measured metabolite/cytokine concentrations throughout the progression of the infection and to better distinguish those with severe COVID-19 who will recover and those with higher risk of death. There is also limited data regarding if organ failure was due to COVID-19 or due to pre-existing chronic disease and worsened by COVID-19. We also recognize the lack of a control group classified as septic patients due to a viral confirmed disease. However, we provide here a comparison of the metabolic pattern associated to sepsis reported in previously published studies for sepsis caused by viral or bacterial agents others than SARS-CoV-2.

## Supporting information

**S1 Fig. Correlation plots between metabolites and cytokines in mild (G2: Blue) and critically ill patients (G4: Red).** (A) ADMA vs. TNF-α (B) kynurenine: tryptophan ratio vs. IL-6 (C) kynurenine: tryptophan ratio vs. IP-10 (D) glucose vs. IP-10 (E) tryptophan **vs.** IP-10 (F) phenylalanine vs. IP-10 (G) trans- hydroxyproline vs. IP-10 (H) methionine vs. IP-10 (I) citric acid vs. NLR (J) pyruvic acid vs. NLR (K) lactic acid vs. NLR (L) homovallinic acid vs. NLR. Spearman correlations with p-values are shown for each correlation.
(TIF)

**S2 Fig. Correlation plots between lipids and cytokines in mild (G2: Blue) and critically ill patients (G4: Red).** (A) LysoPC a 20:3 vs. IL-6 (B) LysoPC a 20:4 vs. TNF-α (C) LysoPC a 26:0 vs. TNF-α (D) PC aa 32:2 vs. IFN λ1 (E) PC aa 36:0 vs. IFN λ1 (F) LysoPC a 20:3 vs. IP-10 (G) LysoPC a 14:0 vs. IP-10 (H) LysoPC a 16:0 vs. IP-10 (I) LysoPC a 16:0 vs. NLR (J) LysoPC a 14:0 vs. NLR (K) LysoPC a 18:2 vs. NLR (L) C4 vs. NLR. Spearman correlation coefficients with p-values are shown for each correlation.
(TIF)

**S3 Fig. Multivariate analysis of non-survivors vs. survivors.** (A) Multivariate analysis from plasma metabolome profile of non-survivors vs. survivors from G4 Score scatter plot based on the PLS-DA models to explain the mortality (black: non-survivors, green: survivors). (B) rank of the top 20 metabolites identified by PLS-DA using VIP score on x- axis. The most discriminating metabolites are shown by descending coefficient scores coefficients. The color boxes indicate increased (red) or decreased (blue) metabolite concentrations.
(TIF)

**S4 Fig. Multivariate and univariate analyses from plasma metabolome profile of COVID-19 patients with and without obesity and diabetes.** (A) Score scatter plot based on PLS-DA models to explain the diagnosis (COVID-WDO: green, COVID-DO: red). (B) 10-cross validation of PLS-DA. (C) Univariate analysis showing the four significant metabolites. The color boxes indicate whether metabolite concentration was increased (red) or decreased (green). Figures were produced in MetaboAnalyst software v 4.0 (https://www.metaboanalyst.ca/).
(TIF)

**S1 Table. Clinical and demographic characteristics of survivors and non-survivors from critically ill group of patients.**
(DOCX)

**S2 Table. Adjusted logistic regression models for diabetes and obesity.**
(DOCX)

## Acknowledgments

To every nurse, assistant, clinician, resident, clinical laboratory scientist and technician from the Mexican Health Services. To Tania Karina Dominguez Covarrubias, Esther Alejandra Mora Medina and Dr. Manuel Reta Hernandez for the project administration. Also, we want to give thanks to Dr. Daniel Yero, for their critical review of the manuscript.

## Author Contributions

**Conceptualization:** David S. Wishart, José A. Enciso-Moreno, Yamilé López-Hernández.

**Formal analysis:** Ana Sofía Herrera-Van Oostdam, Jiamin Zheng, Lun Zhang, Julio César Fernández-Ruiz.

**Funding acquisition:** Yamilé López-Hernández.

**Investigation:** Julio E. Castañeda-Delgado, Juan Carlos Borrego, Elizabeth Soto-Guzmán, Fátima Ochoa-González.

**Methodology:** Juan José Oropeza-Valdez, Flor M. Trejo Medinilla, Jesús Adrián López.

**Project administration:** Yamilé López-Hernández.

**Supervision:** Rupasri Mandal, David S. Wishart.

**Writing – original draft:** Ana Sofía Herrera-Van Oostdam, Yamilé López-Hernández.

**Writing – review & editing:** Julio E. Castañeda-Delgado, Joel Monárrez-Espino, José A. Enciso-Moreno.

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
