## [Decision Letter · Decision Letter 0]

26 Jul 2021

PONE-D-21-16581

Immunometabolic signatures predict risk of progression to sepsis in COVID-19

PLOS ONE

Dear Dr. López-Hernández,

Thank you for submitting your manuscript to PLOS ONE. After careful consideration, we feel that it has merit but does not fully meet PLOS ONE’s publication criteria as it currently stands. Therefore, we invite you to submit a revised version of the manuscript that addresses the points raised during the review process.

It is quite an interesting paper. We suggest the authors to better describe the study population especially regarding comorbidities. It would be interesting, if possible, to make a comparison with patients with sepsis from other etiologies. 

We look forward to receiving your revised manuscript.

Kind regards,

Chiara Lazzeri

Academic Editor

PLOS ONE

Journal Requirements:

2. Please provide additional details regarding participant consent. In the ethics statement in the Methods and online submission information, please ensure that you have specified whether consent was informed. If your study included minors, state whether you obtained consent from parents or guardians.

3. We note that Figure 5 in your submission contain copyrighted images. All PLOS content is published under the Creative Commons Attribution License (CC BY 4.0), which means that the manuscript, images, and Supporting Information files will be freely available online, and any third party is permitted to access, download, copy, distribute, and use these materials in any way, even commercially, with proper attribution. For more information, see our copyright guidelines: http://journals.plos.org/plosone/s/licenses-and-copyright.

a. You may seek permission from the original copyright holder of Figure 5 to publish the content specifically under the CC BY 4.0 license. 

Reviewers' comments:

Reviewer's Responses to Questions

**Comments to the Author**

1. Is the manuscript technically sound, and do the data support the conclusions?

Reviewer #1: Yes

2. Has the statistical analysis been performed appropriately and rigorously? 

Reviewer #1: Yes

3. Have the authors made all data underlying the findings in their manuscript fully available?

Reviewer #1: Yes

4. Is the manuscript presented in an intelligible fashion and written in standard English?

Reviewer #1: Yes

5. Review Comments to the Author

Reviewer #1: Very inreresting manuscript focuding on COVID-19-related sepsis immunometabolism.

A few commnets:

1. Line 81"viral sepsis is crucial to the mechanism of COVID-19 pathogenesis". Maybe change to "crucial for COVID-19 mortality?"

2. Plasma samples obtained within 4 days of symptoms onset. Isn't this interval enough to change the biomarkers? Can a sample obtained on day 1 be compared to a sample obtained on day 4?

3. Why IL-2 was not included in the cytokine assays? It's an important sepsis cytokine.

4. The assessment of comorbidities is not very clear in the manuscript. I'm concerned that patients with diabetes or obesity for example have already altered metabolic profiles and the changes might be due to the chronic metabolic dysfucntion rather than COVID-19. How did the authors address this?

5. The qSOFA score does not predict sepsis as mentioned in line 448. It can be a useful tool for mortality prediction though.

6. The authors propose that the immunometabolic changes in COVID-19 can be attributed to viral sepsis. I'm not sure that they have accounbted for secondary bacterial infections. No mention of blood cultures in the text.

7. It'd be interesting to make comparisons between COVID-19 sepsis and sepsis from other causes in terms of immunometabolism. That might give valuable information specifically for the pathophysiology of COVID-19 sepsis.

6. PLOS authors have the option to publish the peer review history of their article (what does this mean?). If published, this will include your full peer review and any attached files.

Reviewer #1: No

---

## [Author Response · Author response to Decision Letter 0]

11 Aug 2021

Dear editor: 

We appreciate the reviewer's effort and time dedicated to the careful and thorough review of our work. 

Please find below a point-by-point description of all the changes and suggestions made to the manuscript. The changes made by us are highlighted in the manuscript by using tracked changes. With the insightful and valuable comments from the reviewers, the clarity and quality of the manuscript have increased considerably. 

The time and valuable comments you gave were included to improve the manuscript, and we are happy to consider further ones if the reviewers require them.

COMMENTS FROM JOURNAL EDITOR

Thanks for the comments about the manuscript. As you can see in the point-by-point response, we included a better description of the influence of diabetes and obesity in the metabolome profile of COVID-19 patients. We also included a new table in the discussion section, comparing the principal metabolic and potential immune biomarkers of sepsis caused by different etiologic agents. We consider that these topics will improve the quality of the manuscript.

We checked the PLOS ONE style templates, verifying that the manuscript meets PLOS ONE's style requirements.

We also included a more detailed paragraph about informed consent in the Material and method section. Lines 142-145: "Written informed consent was obtained from all participants. All patients included in this study were informed in writing regarding collecting their samples for research aims and given the right to refuse such uses".

Regarding Figure 5, We present written permission from the copyright holder to publish the figure specifically under the CC BY 4.0 license. This form was uploaded as an "Other" file in the submission.

COMMENTS FROM REVIEWERS

REVIEWER 1

Firstly, we would like to acknowledge you for such a constructive review. We feel that your suggestions and observations have considerably improved the quality of our manuscript.

Question 1: Line 81 "viral sepsis is crucial to the mechanism of COVID-19 pathogenesis". Maybe change to "crucial for COVID-19 mortality?"

ANSWER: Thanks for your observation. The change was done. Line 81 "crucial for COVID-19 and its associated mortality".

Question 2: Plasma samples were obtained within four days of symptoms onset. Isn't this interval enough to change the biomarkers? Can a sample obtained on day one be compared to a sample obtained on day 4?

ANSWER: Thanks for your comment. Certainly, metabolism and immune response are dynamic processes that continuously change inside the organism. Thereby differences may be found when sampling on different days. However, due to the complexity of this emergent disease and the wide window of symptoms, it is complicated to find an adequate number of persons that could be sampled at the same time after symptoms onset. In our study, we only included patients confirmed in the first five days of symptoms onset. According to the Centers for Disease Control and Prevention and a recent report (https://doi.org/10.1016/S0140-6736(20)30566-3), after the onset of illness or symptoms, the median time it can take to start feeling shortness of breath is about five to eight days. It means that, in general, the disease progresses similarly in most of the patients during the first five days. The problem worsens during eight to 12 days when patients start to experience acute respiratory distress syndrome, and of course, metabolic and immune markers may drastically change. 

Question 3: Why IL-2 was not included in the cytokine assays? It is an important sepsis cytokine.

ANSWER: Thanks for the observation. At the moment of experimental design of this study, which was planned early in 2020, a few pieces of evidence were available about the role of cytokines. We choose one of the panels available, making possible the measurement of different immune response components (inflammatory response, antiviral response, neutrophil activation). We initially postulated our hypothesis about the potential role of IL-6 and IL-18 in the disease due to antecedents based on MER-CoV infection (https://doi.org/10.1099/vir.0.055533-0). Also, due to logistic reasons and considering the prolonged delays in resources delivery, we decided to order the LEGENDplex™ Human Inflammation Panel (13-plex) (Biolegend, USA because of its wide use in the evaluation of inflammatory response. In the last year, cumulative evidence has been described on the role of specific cytokines in the immune response against this disease, and pivotal importance to IL-6 and IL-10 in the cytokine storm release (https://doi.org/10.1016/j.heliyon.2021.e06155;
https://doi.org/10.1080/22221751.2020.1770129; ), while other authors have found that the medians of the maximal plasma IL-2, IL-4, tumor necrosis factor-α (TNF-α), and interferon-γ levels were in the normal range (https://doi.org/10.1038/s41423-020-00522-6). However, we will consider this for further research, considering the importance of measuring as many cytokines as possible and their kinetics. 

Question 4: The assessment of comorbidities is not very clear in the manuscript. For example, I am concerned that patients with diabetes or obesity have already altered metabolic profiles, and the changes might be due to chronic metabolic dysfunction rather than COVID-19. How did the authors address this?

ANSWER: Thanks for your observation. The comorbidities assessment is an essential point to consider. Obesity and diabetes were the most significant comorbidities in our study groups. These diseases share a typical metabolic pattern that leads to a chronic inflammatory state that may predispose to COVID-19. We evaluated the metabolic profiles associated with COVID-19 patients with and without diabetes and obesity following the comment. We demonstrated no differentiation between diabetic and obese COVID-19 patients from those COVID-19 patients without these comorbidities. Only four metabolites were detected in the univariate analysis as significantly altered between both groups: glucose, TMAO, lysoPC an 18:0 and C4OH. These metabolites have been previously associated with obesity and diabetes (https://dx.doi.org/10.1007%2Fs11306-019-1553-y;
https://dx.doi.org/10.1371%2Fjournal.pone.0041456;
https://doi.org/10.1038/oby.2009.510). These metabolites are different from those find by us when comparing COVID-19 patients with different disease severity grades.

Other authors have also explored how the presence of these comorbidities influences the metabolomics differences found in COVID-19 patients concerning healthy controls and patients with different severity grades. Shi and cols compared the serum metabolome profiles of COVID-19 patients with and without comorbidities. The results showed that none of the patients with hypertension, diabetes, or fatty liver disease could be distinguished from patients without comorbidities. (https://dx.doi.org/10.1016%2Fj.metabol.2021.154739). Similarly, Marin Corral et al. showed similar results in metabolic pathways when analyzing COVID-19 non-obese and obese patients separately (https://doi.org/10.3390/ijms22094794). These results suggest that the effect of SARS-CoV-2 infection on the metabolome profile is much greater than comorbidities.

We included this section in the main text and a new figure explaining the evaluation of the disease's influence on the metabolome. 

Additionally, the multivariate logistic regression models proposed by us were also adjusted for obesity and diabetes. We included a new supplementary table (Supplementary Table 2. When considering the comorbidities by separate, obesity is a significant risk factor for discriminating between G1 and G4, while diabetes is critical to discriminate between G3 and G4. However, none of these comorbidities significantly contribute to the logistic regression models built with the different panels of metabolites and cytokines. 

Question 5: The qSOFA score does not predict sepsis, as mentioned in line 448. It can be a valuable tool for mortality prediction, in any case.

ANSWER: Thanks for the comment; however, we partially disagree. The Definitions of the Third International Consensus for Sepsis and Septic Shock (Sepsis-3) (Singer et al., 2016) establish that to identify patients with suspected infection and poor prognosis of sepsis, at least two clinical criteria of the score quickSOFA (qSOFA) must be considered, including the respiratory rate of 22 / min or more; mood disturbance; or systolic blood pressure of 100 mm Hg or less.

 qSOFA has also been used to predict mortality and prolonged ICU admission in Emergency Department patients with suspected infection (https://doi.org/10.1016/j.jcrc.2018.08.022). In fact, in the Mexican Institute for Social Security, this parameter is used to evaluate the risk of sepsis. 

Question 6: The authors propose that the immunometabolic changes in COVID-19 can be attributed to viral sepsis. I am not sure that they have accounted for secondary bacterial infections—no mention of blood cultures in the text.

ANSWER: Thanks for the observation. This is an important issue. 

The plasma samples analyzed in the present study were collected within four days of symptoms onset and the first 24 h after hospitalization. Patients were confirmed SARS-CoV-2 positive by qRT-PCR, demonstrating the presence of a viral infection. No other symptoms suggesting bacterial co-infections were detected, at least at the moment of blood sampling. Since the patients arrived at the respiratory TRIAGE with a positive SARS-CoV-2 test, no blood cultures were indicated by protocol upon admission. Only four patients had blood cultures drawn at least two weeks after hospitalization (1.2 % had a positive blood culture), being Klebsiella oxytoca, the microorganism detected. Klebsiella oxytoca is a gram-negative bacterium closely related to Klebsiella pneumonie, causing nosocomial infections mainly in diabetic patients, critically ill patients, or under antimicrobial therapies. In the Material and methods section (lines 146-151), we included the description of how blood cultures were performed. Also, in Table 1, we added a line with the number of positive blood cultures detected. 

In line with our findings, a study from New York City hospitals reported a shallow rate of true bacteremia (1.6%) among COVID-19-positive patients (Sepulveda 2020). In a study done by Yang et al., bacteremia was seen in 3% of cases among non-survivors of COVID-19 patients. In another study, among 267 patients hospitalized with COVID-19 pneumonia, 38 had early blood cultures drawn. No clinically relevant microorganism was isolated from blood, and contaminant microorganisms were recovered in 18% of patients, suggesting no evidence of bacteremia in patients with COVID-19 pneumonia (Haedo 2020). Drake et al. also reported a higher rate of complications primarily driven by non-infectious complications, as the rates of secondary bacterial infection in patients with COVID-19 were lower than described in influenza. The incidence of secondary pulmonary infections reported by Chong et al. was 16% for bacterial infections and lowered for fungal infections (6.3%) in hospitalized COVID-19 patients. 

The Global Sepsis Alliance has stated that SARS-CoV-2 causes sepsis (available at: https://www.global-sepsis-alliance.org/covid19). Since SARS-CoV-2 is an infectious pathogen, it is reasonable to suggest that severe COVID-19 is sepsis that SARS-CoV-2 causes. However, this can be either directly (because it is an infectious agent demonstrated by qRT-PCR) or indirectly due to the damage to tissues, immune system, etc. The increased lymphopenia observed in COVID-19 patients may lead to secondary infections during hospitalization (Wang 2020). Increased lymphopenia was observed in our COVID-19 patients, implying possible damage of lymphocytes by the SARS-CoV-2 virus, as has been previously hypothesized (Zheng 2020). 

We cannot rule out the existence of late bloodstream infections with the present work since we only sampled in the first 24 h after hospitalization, and consequently, the immunometabolic signatures reported by us belong to the first hospitalization hours. Invasive devices, diabetes, glucocorticoid treatment, and a combination of antibiotics are significant predictors of nosocomial infections (Garcia Vidal 2021). Intestinal damage due to SARS-CoV-2 infection, systemic inflammation-induced dysfunction, and IL-6-mediated diffuse vascular damage may increase intestinal permeability and precipitate bacterial translocation (Cardinale 2020). Also, microorganisms, microbial fragments (i.e., LPS), and metabolites (i.e., short-chain fatty acids) may cross the intestinal mucosal barrier and reach the lung, enhancing the susceptibility to secondary pulmonary infections that are predominantly seen in critically ill hospitalized COVID-19 patients.

We included these elements in the discussion section.

Question 7: It would be interesting to compare COVID-19 sepsis and sepsis from other causes in terms of immunometabolism. That might give valuable information expressly for the pathophysiology of COVID-19 sepsis.

Answer: Thanks for the suggestion, it is of great importance. 

We included a new Table (Table 3) in the main text. As shown in Table 3, several works have examined the alterations in metabolite levels associated with sepsis or septic shocks induced by different etiologic agents, such as bacteria and fungi. To date, no single compound has shown sufficient sensitivity and specificity to be used as a routine biomarker for early diagnosis and prognosis of septic shock. In terms of immunometabolism, only a few works have been published (Saric et al, 2010; Mickiewicz et al., 2015), demonstrating that a combination of metabolic and immune biomarkers may improve the identification and the prognosis for sepsis. Validated markers to differentiate between viral or bacterial sepsis have not been developed up to date. In Table 3 we can observe that, in general, metabolic pathways, such as glycolysis, TCA cycle, fatty acid oxidation, and amino acid pathways, play essential roles in sepsis and septic shock associated with different causal agents. 

Kindly,

Dr. Yamilé López Hernández

Dr. José A. Enciso Moreno

Corresponding authors

---

## [Editor Report · Decision Letter 1]

16 Aug 2021

Immunometabolic signatures predict risk of progression to sepsis in COVID-19

PONE-D-21-16581R1

Dear Dr. López-Hernández,

We’re pleased to inform you that your manuscript has been judged scientifically suitable for publication and will be formally accepted for publication once it meets all outstanding technical requirements.

Kind regards,

Chiara Lazzeri

Academic Editor

PLOS ONE
---

## [Editor Report · Acceptance letter]

19 Aug 2021

PONE-D-21-16581R1 

Immunometabolic Signatures Predict Risk of Progression to Sepsis in COVID-19. 

Dear Dr. López-Hernández:

I'm pleased to inform you that your manuscript has been deemed suitable for publication in PLOS ONE. Congratulations! Your manuscript is now with our production department. 

Kind regards, 

on behalf of

Dr. Chiara Lazzeri 

Academic Editor

PLOS ONE